# Osteoarthritis in Pseudoxanthoma Elasticum Patients: An Explorative Imaging Study

**DOI:** 10.3390/jcm9123898

**Published:** 2020-12-01

**Authors:** Willem Paul Gielis, Pim A. de Jong, Jonas W. Bartstra, Wouter Foppen, Wilko Spiering, Annemarie M. den Harder

**Affiliations:** 1University Medical Center Utrecht, Department of Orthopaedics, Utrecht University, 3584 CX Utrecht, The Netherlands; 2University Medical Center Utrecht, Department of Radiology, Utrecht University, 3584 CX Utrecht, The Netherlands; P.deJong-8@umcutrecht.nl (P.A.d.J.); jbartstr@umcutrecht.nl (J.W.B.); W.Foppen@umcutrecht.nl (W.F.); amdenharder@gmail.com (A.M.d.H.); 3University Medical Center Utrecht, Department of Vascular Medicine, Utrecht University, 3584 CX Utrecht, The Netherlands; W.Spiering@umcutrecht.nl

**Keywords:** pseudoxanthoma elasticum, joint pain, osteoarthritis, computed tomography, pyrophosphate

## Abstract

Pseudoxanthoma elasticum (PXE) is a systemic disease affecting the skin, eyes, and cardiovascular system of patients. Cardiovascular disease is associated with osteoarthritis (OA), which is the most common cause of joint pain. There is a lack of systematic investigations on joint manifestations in PXE in the literature. In this explorative study, we aimed to investigate whether patients with PXE are more at risk for developing osseous signs of OA. Patients with PXE and hospital controls with whole-body low-dose CT examinations available were included. OA was assessed using the OsteoArthritis Computed Tomography (OACT)-score, which is a 4-point Likert scale, in the acromioclavicular (AC), glenohumeral (GH), facet, hip, knee, and ankle joints. Additionally, intervertebral disc degeneration was scored. Data were analyzed using ordinal logistic regression adjusted for age, body mass index (BMI), and smoking status. In total, 106 PXE patients (age 56 (48–64), 42% males, BMI 25.3 (22.7–28.2)) and 87 hospital controls (age 55 (43–67), 46% males, BMI 26.0 (22.5–29.2)) were included. PXE patients were more likely to have a higher OA score for the AC joints (OR 2.00 (1.12–3.61)), tibiofemoral joint (OR 2.63 (1.40–5.07)), and patellofemoral joint (2.22 (1.18–4.24)). For the other joints, the prevalence and severity of OA did not differ significantly. This study suggests that patients with PXE are more likely to have structural OA of the knee and AC joints, which needs clinical confirmation in larger groups and further investigation into the mechanism.

## 1. Introduction

Pseudoxanthoma elasticum (PXE) is a systemic disease affecting the skin, eyes, and cardiovascular system of patients. It is caused by mutations in the *ABCC6* gene and it is associated with low levels of inorganic pyrophosphate (PPi) [1]. PPi is found in bones and inhibits the precipitation and dissolution of hydroxyapatite (HA). Therefore, it is thought to regulate the entry and exit of calcium and phosphate in mineralized tissues and stabilize already formed calcifications [2,3]. In soft tissues, PPi is an important inhibitor of calcification. A deficiency of PPi is characterized by extensive arterial calcifications. The consequences of a deficiency in the PPi homeostasis are shown in several monogenetic disorders. In Generalized Arterial Calcification of Infancy syndrome (*GACI, OMIM #208000*), the complete lack of PPi results in extensive arterial calcification at birth, but calcification of the joints has also been described in these children [4]. Arterial calcification due to a deficiency in *CD73* (*ACDC, OMIM #211800*) typically results in both periarticular and arterial calcification due to the increased conversion of PPi into calcification promoting phosphate [5,6,7]. The evaluation of the joints in PXE is limited to a recent study on peri-articular calcifications around the shoulder [8].

In our PXE practice, we noticed that several PXE patients complained about painful joints. In some case reports, PXE is co-existing with Rheumatoid arthritis, cervical arthritis, and Still’s disease, but we did not find systematic investigations into structural joint disease in PXE [9,10,11]. Osteoarthritis (OA) is the main cause of disability and joint pain in the general population [12]. Furthermore, OA is related to cardiovascular disease. [13] The most important risk factors are age, body mass index (BMI), gender, and occupation [14,15]. OA is a multifactorial disease with a clear genetic component, but “treatable” phenotypes have not been discovered [16]. A promising potential treatment target is the remodeling of the subchondral bone [17,18].

The goal of the current study was to investigate whether PXE is associated with a higher prevalence of OA-related structural bone disease compared to hospital controls. This line of research could give insight into the origin of the joint pain in PXE patients.

## 2. Materials and Methods

### 2.1. Patients

To determine the prevalence of OA in PXE patients, we used a cohort of consecutive patients with a confirmed PXE diagnosis [19] from the UMC Utrecht in the Netherlands. At least two of the three diagnostic criteria should be fulfilled: skin lesions; peau d’orange or angioid streaks; pathogenic variants on both *ABCC6* alleles. [19] Sanger sequencing was performed to identify single nucleotide polymorphisms (SNPs) and small deletions and insertions, and multiplex ligation-dependent probe amplification (MLPA) was performed to screen for larger deletions in the *ABCC6* gene (reference sequence NM_001171.5, MLPA kit P092B (MRC Holland, Amsterdam, the Netherlands) [20]. All patients with PXE received a non-contrast-enhanced whole-body Computed Tomography (CT) as part of the routine clinical workup to evaluate the amount of vascular calcification. CT acquisitions were low dose (effective dose <3 mSv in a 70 kg adult male) and performed on a 64-slice CT system (Brilliance 64, Philips). PXE Patients did not receive bisphosphonate or anti-vitamin K treatment at the time of or before the scan. 

We used a cohort of hospital controls to compare the prevalence of OA. These patients received a whole-body low-dose CT as part of a fluorodeoxyglucose (FDG) positron emission tomography (PET) CT examination between June 2011 and November 2015 for various medical indications. These scans were performed on a Siemens Biograph 40 scanner (Siemens Healthcare, Erlangen, Germany). Patients with suspicion of endocarditis, vasculitis, osteomyelitis, arthritis, or infected osteosynthesis material were excluded. Of all patients, age, gender, BMI, and smoking status within 6 months of the CT acquisition were extracted from the electronic patient file. Previously, Kranenburg et al. investigated the prevalence and severity of arterial calcifications in the same cohort, which is described in detail elsewhere [21].

### 2.2. Ethical Approval

The need for informed consent was waived by the local institutional review board of the UMC Utrecht (protocol number 15/446-C), since this concerned a retrospective analysis of data acquired in routine clinical care where clinical and radiological data as acquired in routine clinical care were provided in an anonymized fashion to the researchers.

### 2.3. Scoring

OA was assessed on the whole-body CTs using the OsteoArthritis Computed Tomography (OACT)-score. [22] One experienced observer (WPG) scored all scans in random order. His intra-observer and inter-observer reliability compared to two other trained readers has been reported previously [22]. Only CT images were assessed, without Positron Emission Tomography (PET) data or Digital Imaging and Communications in Medicine (DICOM) tags. The acromioclavicular (AC) joints, glenohumeral (GH), hip, knee, and ankle joints were scored using a 4-grade scale with a score of 0 meaning no OA and a score of 3 meaning the most advanced stage of OA (osteophytes, marked joint space narrowing, and subchondral sclerosis/cysts). Joints with a prosthesis received the highest score possible (score 3). Degenerative disc disease and facet joint OA were scored separately for the cervical, thoracic, and lumbar levels. For each section, only the scores for the two most degenerated levels were given a 4-grade scale based on disc/joint space narrowing, osteophytes, sclerosis and for degenerative disc disease specifically; endplate irregularity. An example of the various grades in the tibiofemoral joint is presented in Figure 1.

### 2.4. Statistical Analysis

Statistical analysis was performed with RStudio version 1.1.414 (RStudio Team, Boston, MA, USA). Normally distributed continuous variables are provided as mean ± standard deviation and non-normally distributed continuous variables as median (interquartiles). Categorical variables are provided as n (percentage). Multiple imputation (number of imputations = 25) was used for missing data (R package ‘mice’, based on classification and regression trees). Overall, 2% of the data was missing. The imputations of the missing variables were merged into a single variable by computing the mean of all imputed values (R package ‘sjmisc’). Therefore, the analysis was performed on a single dataset, which is a suitable method when the proportion of missing data is limited [23]. Ordinal logistic regression was applied (R package ‘MASS’) with the OA score for each joint as an outcome variable. Unadjusted and adjusted ordinal logistic regression was performed with age, gender, BMI, smoking status (current smoker yes or no), and group (PXE or hospital control) as dependent variables. The chi-squared score test for the proportional odds assumption was used to assess whether the main model assumption was violated or not (R package ‘VGAM’). Reported are the crude and adjusted proportional Odds Ratios (ORs) with 95% confidence intervals (CI). Sensitivity analysis was performed for the hip and the knee after excluding patients with a joint prosthesis. In joints that showed a higher prevalence of OA in the PXE group, a subgroup analysis was performed to test the association between the genotype and the prevalence of OA. The PXE group was stratified into the number of truncating variants of the *ABBC6* gene: 2 truncating; 1 truncating and 1 non-truncating; 2 truncating variants. A similar ordinal logistic regression model was used to test the association between the number of truncating mutations and the prevalence of OA.

## 3. Results

In total, 106 patients with PXE and 87 hospital controls were included. Baseline characteristics are provided in Table 1. The median age was 56 (48–64) years in PXE patients and 55 (43–67) years in hospital controls. Almost half of the patients were male (42% of PXE patients; 46% of hospital controls) and the median BMI was 25.3 (22.7–28.2) in PXE patients and 26.0 (22.5–29.2) in hospital controls. The indication for the FDG PET-CT in the hospital controls was suspicion of infection (*n* = 53, 61%), malignancy (*n* = 33, 38%), or lymphadenopathy (*n* = 1, 1%). Overall, 2% of the data was missing with a maximum of 11 (12%) missing scores per variable for the ankle. The number of missings per variable is provided in Appendix A. Missing scores were caused by poor image quality or the lack of coverage of a joint in the field of view. Several patients in the control group had a joint prosthesis of the hip (five prostheses in four patients) or knee (three prostheses in two patients). We could stratify 98 patients in the PXE group based on genetic information. Five patients had two non-truncating gene variants, 32 patients had one truncating and one non-truncating variants, and 61 patients had two truncating variants of the *ABCC6* gene (Appendix A). The scores for each joint are provided in Figure 2. The results of the ordinal logistic regression are shown in Table 2. 

The test for the proportional odds assumption was always non-significant, and therefore, the assumptions for performing ordinal logistic regression were not violated. After adjustment, PXE patients were more likely to have a higher OA score for the AC joints (OR 2.00 (1.12–3.61)), tibiofemoral joint (OR 2.63 (1.40–5.07)), and patellofemoral joint (OR 2.22 (1.18–4.24)). Sensitivity analysis after excluding joints with a prosthesis showed the same direction and effect size. No differences were found between males and females. Compared with patients with two truncating variants, OA scores for the AC (OR 0.154 (0.033–0.712)), and patellofemoral joint (OR 0.137 (0.025–0.739)) were lower in patients with two non-truncating variants (Appendix A). Compared with patients with two truncating variants, OA scores for the tibiofemoral joint (OR 0.407 (0.208–0.797)) were lower in patients with one non-truncating variant and one truncating variant. 

## 4. Discussion

In this case-control study, we found that patients with PXE were more likely to have osseous signs of OA in the knee and AC joints compared to hospital controls. Our clinicians noted that PXE patients complained of painful joints more often than expected. This explorative study suggests that the patients may develop OA earlier than expected. In addition, two truncating variants appeared to be associated with a higher prevalence of OA in knees and acromioclavicular joints. However, our observations require prospective validation preferable by clinical assessment and (magnetic resonance) imaging. If confirmed, further investigation into the pathogenesis and possible treatment is needed.

What could be a possible mechanism? PPi is present at both intracellular and extracellular levels. At physiological levels, PPi suppresses HA crystal formation and is a precursor of inorganic phosphate (Pi). In normal physiology, the Pi/PPi ratio is precisely balanced to prevent pathologic calcification. A disbalance in this ratio results in a calcification-promoting environment, where low PPi levels result in HA formation, whereas excess PPi results in the formation of calcium pyrophosphate dehydrate (CPPD) crystal deposition [24]. Studies in *ENPP1*-null mice and *ank/ank* mice, resulting in low PPi levels, show extensive HA crystal deposits in articular cartilage [25,26,27]. Although patients with PXE have a different mutation, namely in the *ABCC6* gene, it also results in low PPi blood levels. Bisphosphonates are nonhydrolyzable PPi analogs. A recent randomized controlled trial showed that a bisphosphonate can limit the development of arterial calcification in patients with PXE [28]. Some studies suggest that bisphosphonates are effective in subgroups of patients with OA, specifically in patients with active remodeling of the subchondral bone [17,29]. Whether this effect is mediated through PPi levels or due to other factors such as the suppression of bone turnover is unclear. In future research, it should be determined whether the increased OA seen in the current study is attributable to HA crystal formation, and it needs to be confirmed that PXE patients have low PPi levels in synovial fluid. 

Another hypothetical mechanism could be related to vascular disease. The incidence of arterial calcifications around knees and hips is related to the incidence of OA in the same joint, although this relationship is only described in women [30]. Previous research in the present population demonstrated that PXE patients have an increased prevalence of vascular calcifications in arm (20% vs. 3%), femoro-popliteal (74% vs. 44%), and sub-popliteal arteries (84% vs. 38%), but not in the vertebral (17% vs. 10%) and external iliac arteries (16% vs. 30%) when compared to hospital controls [21]. This would explain that the relationship between PXE and OA in the AC and knee joints but not in the hip and spine. As glenohumeral and ankle OA have a low prevalence, we had little power to detect a difference between groups in the present study. Possibly, vascular disease at a local level is pathophysiological related to OA in PXE patients. For future research, it would be interesting to study this effect using mediation analysis. 

Other reasons are also possible. Calcific enthesophytes are related to tendinopathy but also seen in abundance in PXE patients [8]. Tendinopathy is associated with OA and may be an important mediator for OA in PXE patients [31]. Additionally, it has been suggested that patients with PXE have lower levels of vitamin K [32], while low vitamin K levels are also associated with an increased prevalence of OA of the hand and knee [33]. Future research should show whether arterial calcifications, enthesophytes, and low vitamin K levels are a risk factor for OA development in PXE patients. 

This study has several limitations. First, due to the retrospective nature of the study, we did not have clinical data on the symptoms, such as pain and swelling. Future studies should confirm if patients with PXE have more symptomatic OA compared to controls. Second, the patients included in the current study were relatively young with a median age of 56 years, while the incidence of OA is strongly age-related. Therefore, most patients presented with a low OA score. Furthermore, the age range in the hospital control group was larger than in the PXE group, although we corrected for age in the regression analysis. Third, we used CT scans from two different vendors, but the resolution and image quality were similar. We believe this did not influence the OA scores. The disease status was not visible for the reader and PET data, and DICOM tags were not reviewed. However, as CT scanner brands differed for the PXE and control group, full prevention of observer bias was not guaranteed. Fourth, a population-based sample would be the preferable control group. However, we used hospital controls, as whole-body CTs were readily available and thus, harmful radiation for new study participants was avoided. Data on comorbidities were lacking due to limitations posed by the ethical committee to protect the privacy of the participants. Hospital controls most likely have more comorbidities compared to population-based controls [34]. As comorbidities are more frequent in people suffering from OA, our control group might have a higher prevalence of OA compared to population-based controls [35]. However, this would result in a bias toward the null hypothesis, meaning that a stronger association between PXE and OA may be observed when PXE patients are compared with population-based controls. Finally, we did not perform a sample size calculation for the main analyses, as this is a first explorative study. PXE is a rare genetic disorder with an estimated prevalence between 1:25,000 and 1:50,000. This means that there are 350 to 700 PXE patients in the Netherlands [36]. In our opinion, the inclusion and analysis of 106 PXE patients is a substantial number, given the rarity of the disease. When stratifying the PXE patients based on the number of truncating gene variants, only five patients had two non-truncating variants. Therefore, we used the group with two truncating variants as the reference group in our analysis. Differences with the group with two non-truncating variants should be interpreted with care. A larger (multicenter) study including a clinical symptoms score is deemed necessary to truly investigate the OA burden in PXE.

## 5. Conclusions

This study suggests that patients with PXE are more likely to have structural OA of the knee and AC joints. Several explanations for this phenomenon were discussed that may provide new insights for future research in patients with PXE and patients with OA in general. 

## Figures and Tables

**Figure 1 jcm-09-03898-f001:**
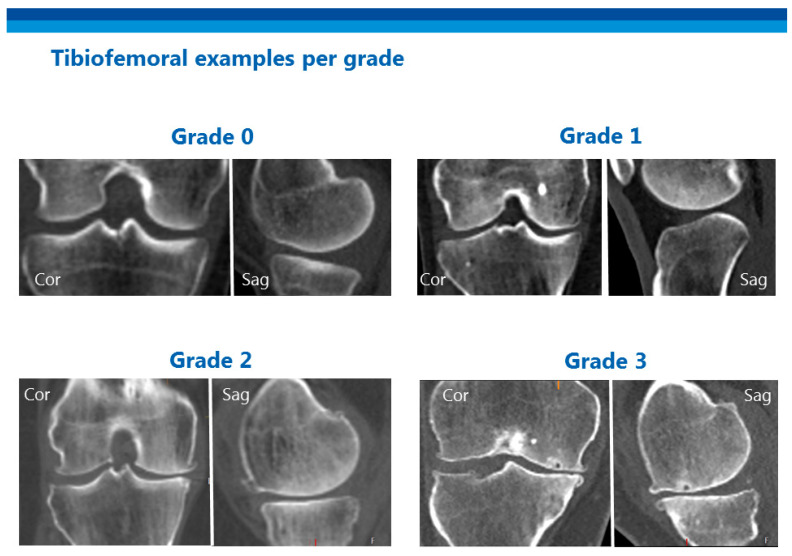
Examples of the different scores of osteoarthritis for the tibiofemoral joint. Sag (Sagittal), Cor (Coronal).

**Figure 2 jcm-09-03898-f002:**
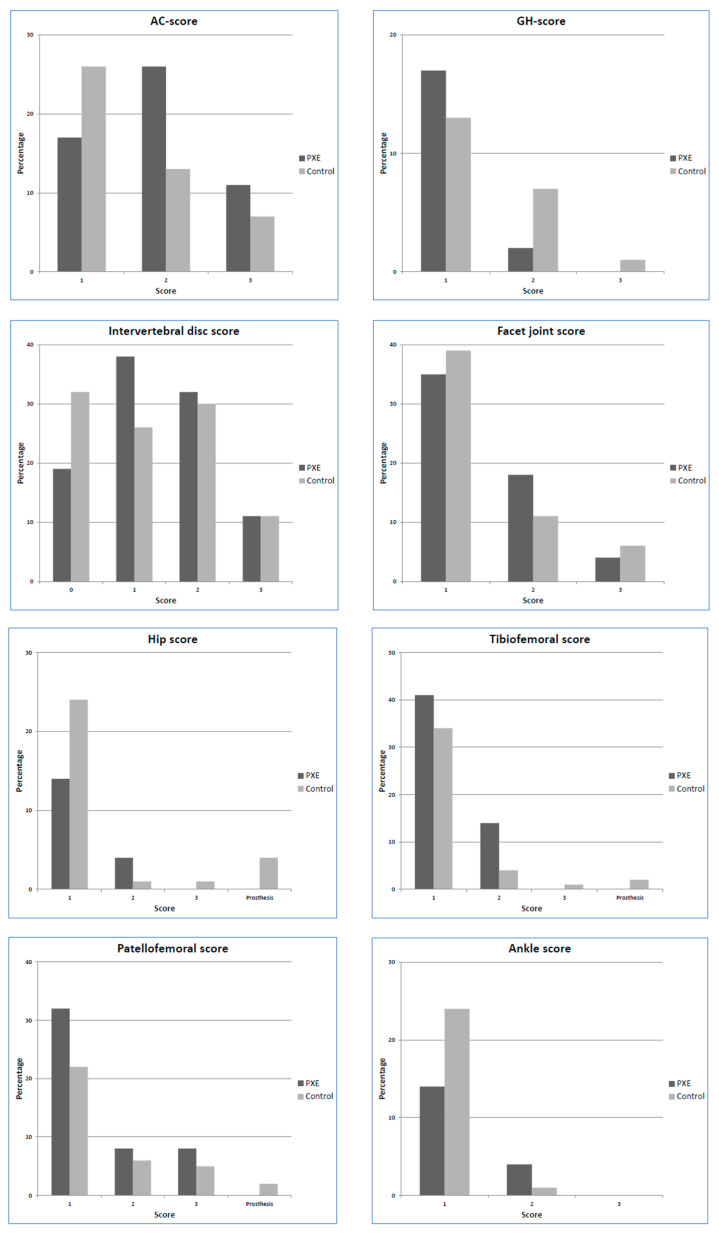
Frequencies of osteoarthritis scores for each joint for the pseudoxanthoma elasticum (PXE) and controls. AC: acromioclavicular; GH: glenohumeral.

**Table 1 jcm-09-03898-t001:** Baseline characteristics.

	Control Group	PXE Group
	*n* = 87	*n* = 106
Age (years)	55 (43–67)	56 (48–64)
Gender (male)	40 (46%)	44 (42%)
BMI (kg/m^2^) *	26.0 (22.5–29.2)	25.3 (22.7–28.2)
Current smoking **	16 (20%)	17 (16%)
Joint prosthesis hip	4 (5%)	0 (0%)
Joint prosthesis knee	2 (2%)	0 (0%)

Data are provided as median (interquartiles) and *n* (%). BMI = body mass index. * missing in seven patients, ** missing in nine patients.

**Table 2 jcm-09-03898-t002:** Results of ordinal logistic regression analysis with each joint as outcome variable.

	Crude OR (95% CI)	*p*-Value	Adjusted OR (95% CI)	*p*-Value
**AC score**	1.69 (0.99–2.90)	0.054	2.00 (1.12–3.61)	0.020
**GH score**	0.83 (0.41–1.70)	0.606	1.19 (0.49–2.54)	0.822
**Intervertebral disc score**	1.34 (0.80–2.26)	0.265	1.61 (0.91–2.87)	0.102
**Facet joint score**	1.08 (0.65–1.83)	0.782	1.31 (0.73–2.38)	0.369
**Hip score**	0.51 (0.26–1.00)	0.050	0.56 (0.27–1.15)	0.117
**Tibiofemoral score**	1.83 (1.06–3.22)	0.033	2.63 (1.40–5.07)	0.003
**Patellofemoral score**	1.70 (0.97–3.02)	0.066	2.22 (1.18–4.24)	0.014
**Ankle score**	0.68 (0.34–1.35)	0.267	0.78 (0.37–1.66)	0.522

Crude analysis and analysis adjusted for age, gender, BMI, and smoking status (current smoker yes or no) were performed. Provided are the results for the group as Odds Ratio (OR) with 95% confidence interval (CI). The hospital controls were used as a reference; therefore, an OR > 1 implies that PXE patients have a higher OA score compared to hospital controls. Patients with a joint prosthesis received the highest score possible. AC, Acromioclavicular; GH, Glenohumeral; OR, Odds Ratio; CI, Confidence Interval.

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
