# Peer review of "Osteoarthritis in Pseudoxanthoma Elasticum Patients: An Explorative Imaging Study"

_jcm, 2020, doi:10.3390/jcm9123898_

Round 1

Reviewer 1 Report

Gielis and colleagues report their experience on osteoarthritis in pseudoxanthoma elasticum (PXE) patients. They reviewed articular changes on whole-body CT scans performed for arterial investigations in a large PXE cohort. The rationale of the study is sound: the Authors noticed that PXE patients often complain of articular pain. This is an original investigation that may be useful for physicians interested in PXE or for rheumatologists. Therefore, this work may be relevant for the readers of JCM.

I have some major comments:

  • Basically, as acknowledged by the Authors, they cannot say that the radiologist is blind to the scans since images are note provided by the same machine
  • The Authors must describe their controls further and insure that they may be used are indisputable controls
  • Discussion / Interpretation of the findings: how the Authors do interpret that some joints could be affected by PXE-related OA and not others?

On a lighter note, some minor changes are needed:

  • There are many typos within the manuscript (even in the title: "joint painT"!). Please re-read all the MS carefully, and correct the word "Blinded" that shows up in several instances (meaning?)
  • The genes need to be written in italics
  • The sentence "Anecdotally we had the impression that PXE patients complained of painful joints more often than expected." is not scientific and needs to be re-written or deleted

Author Response

Thank you for reviewing our manuscript and giving us the opportunity to submit a revision of our manuscript ‘Osteoarthritis and joint pain in PXE patients: an explorative imaging
study
’ (jcm-980462). We kindly thank the reviewers for their time and efforts reviewing the manuscript and providing valuable comments in order to improve our manuscript.

In summary, the reviewers compliment us with a well written and concise manuscript that may be useful for physicians interested in PXE or for rheumatologists. The main concerns were regarding the control group and the use of a different CT scanner for the patient and control groups.

On the next pages you will find our reply to each of the reviewers’ comments. Major changes in text are provided in our reply and changes in the manuscript are marked using the ‘Track Changes’ tool in MS word.

Please do not hesitate to contact us in case of any additional comments.

On behalf of all authors, yours sincerely,

Willem Paul Gielis, MD

Reviewer 1

Gielis and colleagues report their experience on osteoarthritis in pseudoxanthoma elasticum (PXE) patients. They reviewed articular changes on whole-body CT scans performed for arterial investigations in a large PXE cohort. The rationale of the study is sound: the Authors noticed that PXE patients often complain of articular pain. This is an original investigation that may be useful for physicians interested in PXE or for rheumatologists. Therefore, this work may be relevant for the readers of JCM.

Dear reviewer, thank you for the time invested in reading of our manuscript and your valuable comments. Please see our point-by-point responses below.

I have some major comments:

Basically, as acknowledged by the Authors, they cannot say that the radiologist is blind to the scans since images are note provided by the same machine

Authors response: Reviewer 3 also posed issues on the use of two different scanners in the present study. We combined both issues into one response and action.

The reader was blind to disease status, but indeed two different machines were used. Important information was missing from our methods, namely that only the CT images from the PET-CT was read, PET data or DICOM tags were not assessed. It would be very difficult to judge the scanner type based on the images alone.

Author action:

Rewritten text in methods: One experienced observer (WPG), scored all scans in random order. His whom intra-observer reliability and inter-observer reliability compared to two other trained readers, is reported previously, scored all scans blinded to the disease status. [20] Only CT images were assessed, without PET data or DICOM tags.

Rewritten text in discussion to: “The disease status was not visible for the reader and PET data and DICOM tags were not reviewed. However, as CT scanner brands differed for the PXE and control group, full prevention of observer bias was not guaranteed.”

The Authors must describe their controls further and ensure that they may be used are indisputable controls

Author response:

The control group was defined based on the date and indications of the scans. Within the ethical conditions of the Ethical approval we were allowed to retrieve basic information from the electronic patient files. However, this remains limited to the baseline characteristics as described. The control group will always be disputable. We believe the preferable control group would have been population-based controls. However, it would expose participants to harmful radiation. Hospital controls will most likely have more comorbities compared to population-based controls. As comorbidities are related to a higher prevalence of OA, this would mean that the differences between PXE patients and our control group are most likely smaller compared to PXE patients and population-based controls.

Author action: re-written text in discussion:

“Fourth, a population-based sample would be the preferable control group. However, we used hospital controls, as whole body CTs were readily available and harmful radiation for new study participants was thus avoided. Data are more frequent in people suffering from OA, our control group might have a higher prevalence of OA compared to population-based controls [35]. This would however result in a bias towards the null hypothesis, meaning a stronger association between PXE and OA may be observed when PXE patients are compared with population-based controls.

Discussion / Interpretation of the findings: how the Authors do interpret that some joints could be affected by PXE-related OA and not others?

Author response: We thank you for pointing out that we did not add this valuable issue into our discussion.

Author action: added interpretation in the discussion: “PXE patients have an increased prevalence of vascular calcifications in arm, femora-popliteal and sub-popliteal arteries, but not in the arteries around the spine and hip [19]. The incidence of arterial calcifications around knees and hips is related to the incidence of OA in the same joint, although this relationship is only described in women [28]. The arm, femora-popliteal and sub-popliteal arteries supply the shoulder, knee and ankle joints. This would explain that found a relationship between PXE and OA in the AC and knee joints, but not in the hip and spine. As glenohumeral and ankle OA have a low prevalence we had little power to detect a difference between groups in the present study. It is possible that vascular disease at a local level is pathophysiological related to OA in PXE patients. For future research, it would be interesting to study this effect using mediation analysis.”

On a lighter note, some minor changes are needed:

There are many typos within the manuscript (even in the title: "joint painT"!). Please re-read all the MS carefully, and correct the word "Blinded" that shows up in several instances (meaning?)

Author action: We reread the manuscript carefully and replaced blinded by the institution name.

The genes need to be written in italics

Author action: applied italics to genes

The sentence "Anecdotally we had the impression that PXE patients complained of painful joints more often than expected." is not scientific and needs to be re-written or deleted

Author action: sentence changed to: “Our clinicians noted that PXE patients complained of painful joints more often than expected.”

Reviewer 2 Report

Even if the manuscripts deals with a rare disease, my opinion is that the topic is interesting and the manuscript is very well written, concise and precise. It scientifically sounds, materials and results are well explained and it has been well conducted. I think it is suitable for this journal (just at line 1: please use extensive form for the first use of an acronym, in this case, PXE).

Author Response

Thank you for reviewing our manuscript and giving us the opportunity to submit a revision of our manuscript ‘Osteoarthritis and joint pain in PXE patients: an explorative imaging
study
’ (jcm-980462). We kindly thank the reviewers for their time and efforts reviewing the manuscript and providing valuable comments in order to improve our manuscript.

In summary, the reviewers compliment us with a well written and concise manuscript that may be useful for physicians interested in PXE or for rheumatologists. The main concerns were regarding the control group and the use of a different CT scanner for the patient and control groups.

On the next pages you will find our reply to each of the reviewers’ comments. Major changes in text are provided in our reply and changes in the manuscript are marked using the ‘Track Changes’ tool in MS word.

Please do not hesitate to contact us in case of any additional comments.

On behalf of all authors, yours sincerely,

Willem Paul Gielis, MD

Reviewer 2

Even if the manuscripts deals with a rare disease, my opinion is that the topic is interesting and the manuscript is very well written, concise and precise. It scientifically sounds, materials and results are well explained and it has been well conducted. I think it is suitable for this journal (just at line 1: please use extensive form for the first use of an acronym, in this case, PXE).

Author response: Dear reviewer, thank you for the time invested in reading our work and praising comments.

Author action: Applied extensive form in the title.

Reviewer 3 Report

Gielis et al. reports an exploratory study on Osteoarthritis and joint paint in PXE patients.

Although the question is relevant, this study raises a number of questions and remarks :

  • The title mentions the notion of joint paint and there is no solid basis for this. Indeed, in the article, the authors mention Anecdotally we had the impression that PXE patients complained of painful joints more often than expected. There is no solid basis for this statement and it should be removed from the title. We do not know how the authors assessed pain. Pain Scale? What about pain in the control group?

  • The abstract also needs to be rewritten at the beginning: the authors start by introducing the subject by the fact that PXE patients complained of joint pain. As this is based on an impression, the authors should start the abstract by introducing the definition of PXE and then say what exists in the literature about osteoarticular damage and then introduce their subject.

  • In the introduction, the paragraph line 39-51 should be put at the beginning of the introduction. Then introduce the subject of joint damage and the aim of the study.

  • The diagnostic criteria for PXE should be mentioned in the material and methods section. The genotype of each patient should be mentioned in an additional table in order to be certain of the diagnosis of PXE related to the ABCC6 mutation. As CD73 CDAD is a PXE like syndrome resulting in joint and vascular damage, it is important to have the genotype of the patients to be able to state that the osteoarticular damage is related to PXE. Especially as the patients recruited in this article are from a cohort of vascular PXE patients.

  • The treatment of the patients should be mentioned as the bisphosphonates used in Holland according to the results of the SPECT study. This should be taken into account in the statistical analyses. Were there patients on anti-vitamin K?

  • Do you have the PPi and vitamin K levels of each patient?

  • The authors mentioned : “Another hypothetical mechanism could be related to vascular disease. PXE patients have an increased prevalence of vascular calcifications [19]. The incidence of arterial calcifications around 161 knees and hips is related to the incidence of OA in the same joint, although this relationship is only described in women [28]. It is possible that vascular disease at a local level is pathophysiological related to OA in PXE patients.” The authors have to weigh this up because two recent studies have found no calcifications in the popliteal artery (DOI: 10.1371/journal.pone.0096003; doi.org/10.3390/jcm9113448). It would be interesting if the authors could give the calcium score at the popliteal artery level in order to support their argumentation, even if this was not an objective of the article. But as mentioned, and that the calcium score is easily measurable at the level of the popliteal artery on CT-Scan, the authors could give this information to support their argument.

  • This study was carried out on two different machines, one of which was dedicated to PET-CT. This constitutes a major bias in the analysis.

  • Only one experienced observer reread the scanners. It would be interesting to have two independent and blind observers to have inter-observer reproducibility and to draw robust conclusions.

Author Response

Thank you for reviewing our manuscript and giving us the opportunity to submit a revision of our manuscript ‘Osteoarthritis and joint pain in PXE patients: an explorative imaging
study
’ (jcm-980462). We kindly thank the reviewers for their time and efforts reviewing the manuscript and providing valuable comments in order to improve our manuscript.

In summary, the reviewers compliment us with a well written and concise manuscript that may be useful for physicians interested in PXE or for rheumatologists. The main concerns were regarding the control group and the use of a different CT scanner for the patient and control groups.

On the next pages you will find our reply to each of the reviewers’ comments. Major changes in text are provided in our reply and changes in the manuscript are marked using the ‘Track Changes’ tool in MS word.

Please do not hesitate to contact us in case of any additional comments.

On behalf of all authors, yours sincerely,

Willem Paul Gielis, MD

Reviewer 3

Gielis et al. reports an exploratory study on Osteoarthritis and joint paint in PXE patients.

Dear reviewer, thank you for the time invested in reading our work and the valuable comments. Please see our point-by-point responses below.

Although the question is relevant, this study raises a number of questions and remarks :

The title mentions the notion of joint paint and there is no solid basis for this. Indeed, in the article, the authors mention Anecdotally we had the impression that PXE patients complained of painful joints more often than expected. There is no solid basis for this statement and it should be removed from the title. We do not know how the authors assessed pain. Pain Scale? What about pain in the control group?

Author response: Pain was not assessed on a standardized basis in the study population. The available CT scans enabled us to study the prevalence of structural features of osteoarthritis in PXE patients and compare that to hospital group. As discussed, future studies are needed to confirm the presences of clinical symptoms.

Author action: We removed joint pain from the title. 

The abstract also needs to be rewritten at the beginning: the authors start by introducing the subject by the fact that PXE patients complained of joint pain. As this is based on an impression, the authors should start the abstract by introducing the definition of PXE and then say what exists in the literature about osteoarticular damage and then introduce their subject.

Author response: The present research question arose naturally from our clinics. However, we agree that the abstract can be reordered to a more conventional set-up. Furthermore, it should briefly discuss the previous literature on joint manifestations of PXE.

Author action: rewritten introduction of the abstract:

“Pseudoxanthoma elasticum (PXE), is a systemic disease affecting the skin, eyes, and cardiovascular system of patients. Cardiovascular disease is associated with osteoarthritis (OA), the most common cause of joint pain. Systematic investigations on joint manifestations in PXE lack from literature. In this explorative study, we aimed to investigate whether patients with PXE are more at risk for developing osseous signs of OA.

In the introduction, the paragraph line 39-51 should be put at the beginning of the introduction. Then introduce the subject of joint damage and the aim of the study.

Author action: We restructured the introduction accordingly.

The diagnostic criteria for PXE should be mentioned in the material and methods section. The genotype of each patient should be mentioned in an additional table in order to be certain of the diagnosis of PXE related to the ABCC6 mutation. As CD73 CDAD is a PXE like syndrome resulting in joint and vascular damage, it is important to have the genotype of the patients to be able to state that the osteoarticular damage is related to PXE. Especially as the patients recruited in this article are from a cohort of vascular PXE patients.

Author response: It is a valid remark that diagnostic criteria are indispensable information for readers. We used the diagnostic criteria as proposed by Plomp et al. for our population.

Author action: added to methods:

“Two of at least three diagnostic criteria should be fulfilled: skin lesions; peau d’orange or angioid streaks; pathogenic mutation on both ABCC6 genes. [20]”

The treatment of the patients should be mentioned as the bisphosphonates used in Holland according to the results of the SPECT study. This should be taken into account in the statistical analyses. Were there patients on anti-vitamin K?

Author response: Patients did not receive bisphosphonates or anti-vitamin K treatment before or at the time of the scan.

Author action: Added to methods:

“PXE Patients did not receive bisphosphonate or anti-vitamin K treatment at the time of or before the scan.”

Do you have the PPi and vitamin K levels of each patient?

Author response: Vitamin K an Ppi levels were not available for the current population. Please also see our response to the following issue.

The authors mentioned: “Another hypothetical mechanism could be related to vascular disease. PXE patients have an increased prevalence of vascular calcifications [19]. The incidence of arterial calcifications around 161 knees and hips is related to the incidence of OA in the same joint, although this relationship is only described in women [28]. It is possible that vascular disease at a local level is pathophysiological related to OA in PXE patients.” The authors have to weigh this up because two recent studies have found no calcifications in the popliteal artery (DOI: 10.1371/journal.pone.0096003; doi.org/10.3390/jcm9113448). It would be interesting if the authors could give the calcium score at the popliteal artery level in order to support their argumentation, even if this was not an objective of the article. But as mentioned, and that the calcium score is easily measurable at the level of the popliteal artery on CT-Scan, the authors could give this information to support their argument.

Author response: In the present exploratory study we wanted to systematically study the prevalence of OA as a possible joint manifestation in PXE patients, as such studies are lacking from literature. In future research we want to include markers of vascular disease, but also pain, PPi and vitamin-K levels and inflammation related outcome markers. The presence of the vascular calcifications in the study of Kranenburg et al. and the presence of OA have a shared distribution, which we now discuss in the discussion. An analysis to study whether the higher prevalence of OA is caused by arterial calcifications needs a comprehensive mediation analysis. However, we believe such analysis is outside the scope of this first systematic study on degenerative joint disease in PXE.

Author action: rewrote discussion:

“Another hypothetical mechanism could be related to vascular disease. The incidence of arterial calcifications around knees and hips is related to the incidence of OA in the same joint, although this relationship is only described in women [28]. Previous research in the present population demonstrated that PXE patients have an increased prevalence of vascular calcifications in arm (20% vs 3%), femora-popliteal (74% vs 44%) and sub-popliteal arteries (84% vs 38%), but not in the vertebral (17% vs 10%) and external iliac arteries (16% vs 30%), when compared to hospital controls [19]. This would explain that found a relationship between PXE and OA in the AC and knee joints, but not in the hip and spine. As glenohumeral and ankle OA have a low prevalence we had little power to detect a difference between groups in the present study. It is possible that vascular disease at a local level is pathophysiological related to OA in PXE patients. For future research, it would be interesting to study this effect using a mediation analysis.”

This study was carried out on two different machines, one of which was dedicated to PET-CT. This constitutes a major bias in the analysis.

Authors response: Reviewer 1 also posed issues on the use of two different scanners in the present study. We combined both issues into one response and action.

The reader was blind to disease status, but indeed two different machines were used. Important information was missing from our methods, namely that only the CT images from the PET-CT was read, PET data or DICOM tags were not assessed. It would be very difficult to judge the scanner type based on the images alone. Therefore, we believe the use of two different machines does not constitute a major bias. However, prevention of observer bias was not fully guaranteed.

Author action:

Rewritten text in methods:

“One experienced observer (WPG), scored all scans in random order. His intra-observer and inter-observer reliability compared to two other trained readers are reported previously. [22] Only CT images were assessed, without PET data or DICOM tags.”

Rewritten text in discussion to:

“The disease status was not visible for the reader and PET data and DICOM tags were not reviewed. However, as CT scanner brands differed for the PXE and control group, full prevention of observer bias was not guaranteed.”

 Only one experienced observer reread the scanners. It would be interesting to have two independent and blind observers to have inter-observer reproducibility and to draw robust conclusions.

Author response: The reliability of the reader is compared to two other readers in a paper that is currently under review at the Journal of Personalized Medicine.

Author action: rewritten methods:

“One experienced observer (WPG), scored all scans in random order. His intra-observer and inter-observer reliability compared to two other trained readers is reported previously. [22]

Round 2

Reviewer 1 Report

Dear Authors,

I am now happy with the MS. I appreciate your efforts to provide a better rationale to the study, and a more cautious discussion of the findings.

Author Response

Dear reviewer,

Thank you for the time invested in reading and commenting on our manuscript. 

Reviewer 3 Report

Osteoarthritis in Pseudoxanthoma Elasticum patients: an explorative imaging study

Dear Authors,

Thank you for answering my questions and improving the manuscript.

However, I had asked in my previous review to put all the mutations of the patients in this study in supplementary files. This is very important for the readers because osteoarthritis has not been studied very much before. So, I suggest that the authors mention this and see if there is a genotype-phenotype correlation for osteoarthritis or not in the results and discussion sections.

I could only agree if all the mutations are provided in the supplementary files section.    

Yours sincerely

Author Response

Dear Authors,

Thank you for answering my questions and improving the manuscript.

However, I had asked in my previous review to put all the mutations of the patients in this study in supplementary files. This is very important for the readers because osteoarthritis has not been studied very much before. So, I suggest that the authors mention this and see if there is a genotype-phenotype correlation for osteoarthritis or not in the results and discussion sections.

I could only agree if all the mutations are provided in the supplementary files section.    

Yours sincerely

Author response:

Dear reviewer,

Thank you for the time invested into our manuscript on your valuable comments. We now included all mutations of the patients in the study in a supplementary file. Furthermore, we divided the genetic variants into truncating vs non-truncating variants. Subsequently, we performed a subanalysis using this stratification of gene variants. To our surprise, this produced interesting results, which we included into our manuscript. As the group of patients with two non-truncating gene variants consists of only 5 patients, the results regarding this group should be interpreted with care.

Author action:

Added to methods:

“Sanger sequencing was performed to identify single nucleotide polymorphisms (SNPs) and small deletions and insertions, and multiplex ligation-dependent probe amplification (MLPA) was performed to screen for larger deletions in the ABCC6 gene (reference sequence NM_001171.5, MLPA kit P092B ((https://www.mrcholland.com/)).”

“In joints that showed a higher prevalence of OA in the PXE group, a subgroup analysis was performed to test the association between the genotype and the prevalence of OA. The PXE group was stratified into the number of truncating variants of the ABBC6 gene: 2 truncating; 1 truncating and 1 non- truncating; 2 truncating variants. A similar ordinal logistic regression model was used to test the association between the number of truncating mutations and the prevalence of OA.”

Added to results:

“We could stratify 98 patients in the PXE group based on genetic information. Five patients had 2 non-truncating gene variants, 32 patients had one truncating and one non-truncating variants, and 61 patients had two truncating variants of the ABCC6 gene.”

“Compared with patients with 2 truncating variants, OA scores for the AC (OR 0.154 [0.033 – 0.712]) and patellofemoral joint (OR 0.137 [0.025 – 0.739]) were lower in patients with 2 non-truncating variants. Compared with patients with 2 truncating variants, OA scores for the tibiofemoral joint (OR 0.407 [0.208 – 0.797]) were lower in patients with one non-truncating and one truncating variants.”

Added to discussion:

“In addition, 2 truncating variants appeared to be associated with a higher prevalence of OA in knees and acromioclavicular joints. However, our observations require prospective validation preferable by clinical assessment and (magnetic resonance) imaging. If confirmed, further investigation into the pathogenesis and possible treatment is needed.”

“When stratifying the PXE patients based on the number of truncating gene variants, only five patients had two non-truncating variants. Therefore, we used the group with two truncating variants as reference group in our analysis. Differences with the group with two non-truncating variants should be interpreted with care. A larger (multicenter) study including a clinical symptoms score is deemed necessary to truly investigate the OA burden in PXE.”

Added to supplementary files:

Gene mutations per patient.

Table with results of subanalysis.
